# Benchmarking Antimicrobial Peptide Identification with Sequence and Structure Representation

## Abstract

The rapid evolution of drug-resistant (DR) microbial has become a severe issue for human health. Antimicrobial peptides (AMPs) are powerful therapeutic drugs for treating DR microbial, but their clinical application is limited by activity and toxicity. Recently, AI has shown its power in discovering the high-activity AMPs, relying on the database of the AMP's wet-lab activity data. However, the activity data from this database are collected from thousands of papers, with their different wet lab experiments setting on one or few types of DR bacteria, have further limits the development of AI methods for AMP identification. Moreover, recently AlphaFold has revolutionized the field of drug discovery, but how can we benefit from the predicted structure for AMP discovery still remains unknown. To address the above challenges, we make two contributions. **a)** We construct the **DRAMPAtlas 1.0** that contains the training set collected from the public and the testing set from our wet lab experiment. Each AMP sequence is equipped with its 3D structure, activity data, and toxicity, where the activity is about six types of DR bacteria. **b)** We conduct extensive experiments for AMP identification, by modeling the 3D structure as voxels or graphs, in conjugate with its sequence information or solely with the structure or sequence. We have made many interesting findings. We hope that our benchmark and findings can benefit the research community to better design the algorithms for high-activity AMP discovery. All code and data associated with the work will be made publicly available after acceptance.

## 1 Introduction

Antimicrobial peptides (AMPs) are short amino acid (AA) sequences that can kill bacteria, fungi, tumor cells, and viruses with the length typically from 6 to 50 (Fjell et al., 2012; Wan et al., 2024). Although people have been discovering AMPs for over 80 years, fewer than 50 AMPs have been under clinical experiment or approved by the US Food and Drug Administration (FDA) (Chen & Lu, 2020). Drug resistance (DR) has become a serious issue for humans, with an increasing number of deaths of over 1.2 million a year (Murray et al., 2022). The membrane disruption and another mechanism of AMPs (Silva et al., 2020), have given the AMPs the potential to deal with the DR issue of conventional antibiotics. Over the past decades, especially the revolution of deep learning (LeCun et al., 2015), scientists have found new ways to effectively find the potential clinically useful AMPs. Still, there is a long way to find the AMPs with high activity like antibiotics with the low toxin, high stability, etc (Chen & Lu, 2020).

The performance of deep learning-based methods highly relies on the data (Li et al., 2024b; Wan et al., 2024), specifically, the ground truth (GT) value and input data pairs. **a)** For the GT values, in the real of the discovery of AMPs, the wet lab data like minimal inhibitory concentration (MIC) values are collected from papers under different experimental setting (Wan et al., 2024). For example, some AMP experiments are conducted under water while another AMP is conducted under the plasma protein solution (Moretta et al., 2021). Besides, according to the classification of DR bacteria from WHO (Willyard, 2017), there is more than one type of DR bacteria. One AMP may perform differently under different types of DR bacteria (Szymczak et al., 2023). Current studies (de Breij et al., 2018; Das et al., 2021; Ma et al., 2022; Huang et al., 2023; Cao et al., 2023; Pandi et al.,

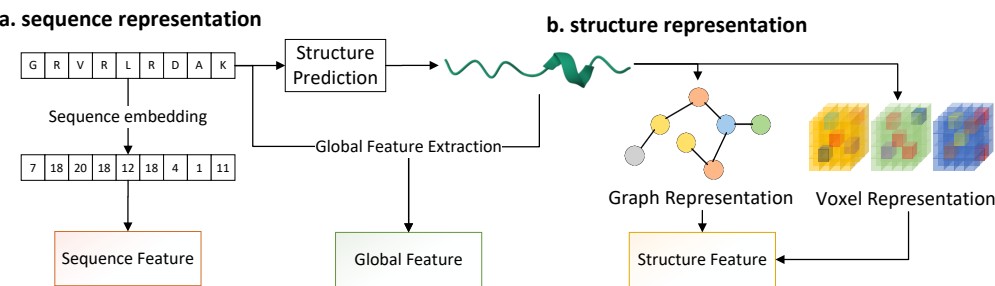

Figure 1: Sequence representation v.s. Structure representation of the AMPs. We aim to uncover the effectiveness of those representation methods in AMP identification.

2023; Szymczak et al., 2023; Li et al., 2024a) only investigate the effectiveness of a few AMPs (usually less than 30) in wet lab under different DR bacteria. **b)** For the input data, current methods for AMPs discover mainly rely on the sequence (Huang et al., 2023; Li et al., 2024a; Cao et al., 2023) except for these works (Wang et al., 2024). Although the Alphafold (Jumper et al., 2021) have revolution the field of drug discovery, the impact of the AMPs discovered with the structure is merely discussed, due to the lake of multi-sequence alignment (MSA) information (Fang et al., 2023). It makes people wonder, how important is the structure information for the AMP discovery. The above situation has been an obstacle in the field of AMP identification. We try to address the above concerns with the following two designs in our benchmark.

1. The intrinsic noise and missing values of different DR bacteria species have been the obstacle for AMP training set construction (Wan et al., 2024). Considering it is unreliable to re-conduct the wet lab experiments for these DR bacteria, we first conduct the wet lab experiments on a family of AMPs, which contains 150 AMPs with their activities on 6 types of several DR bacteria. This dataset is treated as the test set for the evaluation of algorithms. Based on these GT values, we have trained a missing value imputation algorithm (van Buuren & Groothuis-Oudshoorn, 2011) to fill the missing values in the training data set. Thus, we contribute the **DRAMPAtlas 1.0**, with its unique advantage on the clean test set and cleaned training set.

2. The use of structure information in AMP discovery has merely been discussed (Wan et al., 2024). According to the principle that the structure determines the function, the protein structure is quite important (Anfinsen, 1973). Thus, we conduct extensive experiments on the analysis of different structure representations, such as voxels, and graphs, and how other information (secondary structure) from the 3D structure can benefit AMP identification.

With our tailored-constructed **DRAMPAtlas 1.0** in **Section 3**, this work aims to answer the following two questions, one for **feature representation** and another for the **learning representation**: **1. Representation**: How important is structure information & can structure information work together with sequence information? **2. Learning**: Can we benefit from modern neural network structure, feature fusion, or re-balancing strategies?

To answer the above questions, we first set the multi-label classification settings in **Section 4.1** in conjugate with the evaluation metrics. For the first question, we benchmark the input information in **Section 4.2**. We rigorously evaluate the performance of the different types of input data, such as sequence, structure, sequence & structure, etc. We answer the second question in **Section 4.3** by evaluating different types of backbone neural networks and different re-balancing strategies. We also validate the scale-up ability of network parameters. All these trained models will be made available to the public.

Table 1: **Contribution1: A reliable test set dataset for AMP identification.** This dataset is unprecedented with respect to data quality and annotation scales. # indicates the number. "strains" indicates how many types of micro-bacteria are tested in the wet lab. "DS" and "DR" indicate the AMPs are evaluated on drug-sensitive or drug-resistant micro-bacteria, respectively.

| Dataset | Venue | # of seqs | # of species | # of strains | DS/DR |
|---|---|---|---|---|---|
| SAAP (de Breij et al., 2018) | STM 2018 | 25 | 1 | 1 | DS |
| CLaSS (Das et al., 2021) | NBE 2021 | 20 | 2 | 2 | DS |
| SearchAMP (Huang et al., 2023) | NBE 2023 | 130 | 1 | 1 | DS |
| BertAMP (Cao et al., 2023) | BIB 2023 | 40 | 7 | 8 | DS |
| Cell-freeAMP (Pandi et al., 2023) | NC 2023 | 30 | 6 | 6 | 5/1 |
| HydrAMP (Szymczak et al., 2023) | NC 2023 | 26 | 4 | 5 | 3/2 |
| FoundationAMP (Li et al., 2024a) | NC 2024 | 29 | 4 | 5 | 4/1 |
| Ours | - | 151 | 6 | 18 | DR |

## 2 RELATED WORK

### 2.1 REPRESENTATIONS OF ANTIMICROBIAL PEPTIDES (AMPS)

From a global to a local perspective, the features of AMPs can be categorized into global descriptors, sequence descriptors, and structure-based descriptors.

**Global descriptors** summarize properties of peptides using fixed-size vectors, capturing aspects such as sequence composition, structural features, and physicochemical properties (Xu & et al., 2021). These descriptors have been extensively studied (Osorio et al., 2015; van Westen & et al., 2013; Müller et al., 2017; Romero-Molina et al., 2019; Barigye et al., 2021; Chen & et al., 2018); however, employing all available descriptors can lead to high-dimensional and redundant information. Feature-selection algorithms (Saeys et al., 2007) help generate low-dimensional representations better suited for machine learning (ML) models. Constructing global descriptors requires substantial effort and domain knowledge but is useful for capturing specific information when training data is limited. It is worth noting that global features can be derived from both sequence and structural information.

**Sequence-based representations** capture primary amino acid sequences using an $n \times L$ matrix, where $n$ is the number of features per amino acid and $L$ is the sequence length. Amino acids are often represented using one-hot encoding (Chen & et al., 2021), where each amino acid is uniquely identified. However, one-hot encoding does not capture additional amino acid properties, which can be addressed by incorporating physicochemical and evolutionary features (Kawashima, 2000). Deep learning (DL) techniques can learn amino acid embeddings in a data-driven manner (ElAbd & et al., 2020). Sequence-based representations are useful for ML models like recurrent neural networks (Hochreiter & Schmidhuber, 1997; Chung et al., 2014) and have been applied to peptide sequence generation (Wan et al., 2022) and property prediction (Chen & et al., 2021).

For structural representations, there are two prevailing methods: voxel representation and graph representation.

**Voxel representations** involve voxelization of peptides' 3D structures, with each voxel storing information about atom occupancies and properties (Jiménez et al., 2017). Three-dimensional convolutional neural networks (3D-CNNs) (Maturana & Scherer, 2015) process these structures and have been applied to tasks like protein binding site prediction (Jiménez et al., 2017) and protein-ligand binding affinity prediction (Jones & et al., 2021).

**Graph representations** model peptides by using nodes (atoms or residues) and edges (chemical bonds or spatial distances) to form graphs (Wang et al., 2023; Gong et al., 2023). These graphs are encoded using features such as one-hot encodings of atom or residue types and geometric properties. Graph-based inputs are suitable for geometry-related ML tasks and have been used in protein structure prediction (Jumper et al., 2021; Baek et al., 2021), AMP prediction (Yan et al., 2023), molecular conformation generation (Ganea & et al., 2021), and antibody design (Jin et al., 2022).

Table 2: **Contribution2: A structured training set for AMP identification.** This dataset is unprecedented with respect to data quality and annotation scales. # indicates the number. "strains" indicates how many types of micro-bacteria are tested in the wet lab. "DS" and "DR" indicate the AMPs are evaluated on drug-sensitive or drug-resistant micro-bacteria, respectively. Note that our structure is predicted by two models, alphafold and helixfold.

| Dataset | Venue | # of seqs | # of species | # of structure | cleaned |
|---|---|---|---|---|---|
| LAMPv2 (Ye et al., 2020) | 2020 | 23,253 | - | - | No |
| dbAMP 2.0 (Jhong et al., 2022) | 2021 | 26,447 | - | 3444 | No |
| DRAMPv3 Shi et al. (2022) | 2022 | 22,151 | - | 546 | No |
| CAMPr4 (Gawde et al., 2023) | 2022 | 16,945 | - | 933 | No |
| Dbaaspv3 (Pirtskhalava et al., 2021) | 2023 | 22,612 | - | >3,600 | No |
| QLAPD (Wang et al., 2024) | 2024 | 12,914 | 6 | 12,914 | Yes |
| DRAMPAtlas 1.0 | Ours | 23,673 | 6 | 47,346 | Yes |

Although previous studies have employed some of these representation methods for AMP identification, there is still a lack of systematic evaluation of the correlation between different representations and AMP identification results. This gap exists both from the perspective of individual representations and the fusion of multiple representations through neural networks.

## 2.2 LEARNING TO IDENTIFY AMPs

Machine learning-based approaches (García-Jacas et al., 2022; Sidorczuk & et al., 2022) trained on public AMP databases (Pirtskhalava et al., 2021; Waghu et al., 2016; Witten & Witten, 2019; Wang et al., 2016) have been utilized to predict antimicrobial activity from amino acid sequences, a crucial step in AMP development. These approaches employ traditional ML methods like random forests and support vector machines (Meher et al., 2017; Xiao et al., 2013; Fingerhut et al., 2021; Santos-Júnior et al., 2020; Burdukiewicz & et al., 2020; Lawrence & et al., 2021; Bhadra et al., 2018; Pane & et al., 2017), as well as DL-based methods (Yan & et al., 2020; Veltri et al., 2018; Wang et al., 2024). However, variations in input representations, classification strategies for AMPs, and evaluation metrics across studies complicate the determination of the most effective features and models. To ensure fair model comparisons, feature selection and classification of training data require standardized benchmarking (García-Jacas et al., 2022; Sidorczuk & et al., 2022). Improving data quality is also vital, as public AMP databases often contain data from diverse experimental conditions and organisms (Wan et al., 2024). Therefore, in this work, we build a high-quality testing set for AMP identification through extensive wet lab experiments.

## 3 DRAMPATLAS 1.0: BENCHMARKING AMPs IDENTIFICATION

### 3.1 GROUND TRUTH CONSTRUCTION FOR THE TESTING SET

We constructed a multi-label test set through extensive wet lab experiments on a series of peptides. This test set includes the Minimum Inhibitory Concentration (MIC) performance of each peptide against six high-risk Drug-Resistant (DR) bacterial classes listed by WHO. The MIC values for each DR bacterium are represented by the average MICs of three different strains. The specific names of these bacteria are: *A. baumannii*, *P. aeruginosa*, *E. coli*, *K. pneumoniae*, *S. enterica*, and *S. aureus*. All peptides were tested under uniform wet-lab conditions, and each strain was tested twice to obtain an average MIC value as the assessment result. The entire process spanned 18 months and cost over $100,000, with detailed specifics provided in Appendix Table 1. The template peptides, from which these similar peptides were derived, were generated using a conditional variational autoencoder (Wang et al., 2024). Through manual expertise and various methods, 151 peptide sequences were modified from this template.

In Table 1, we present a comparison between our new test set with previous works, highlighting that our method currently involves the largest dataset of peptides verified through extensive wet-lab experiments under uniform experimental conditions for MIC testing. Unlike previous studies (de Breij et al., 2018; Huang et al., 2023) that often focused on a single type of bacteria, such as *Staphylococ-*

*cus aureus*, and were not necessarily targeting drug-resistant (DR) bacteria, our dataset covers six different types of microorganisms and includes 18 different strains, most of which are DR strains. This not only increases the diversity and representativeness of the data but also allows our method to be evaluated in more complex and realistic environments. Furthermore, previous datasets like SAAP (de Breij et al., 2018), CLaSS (Das et al., 2021), and SearchAMP (Huang et al., 2023), despite achieving certain outcomes in their respective areas, have fewer sequences, species diversity, and strain numbers compared to our dataset. For instance, the SAAP dataset only includes 25 sequences and 1 strain, whereas our dataset contains 151 sequences and 18 strains, significantly enhancing the model's generalization ability across diverse samples. Additionally, some datasets only evaluated drug-sensitive (DS) strains, while our dataset encompasses both drug-sensitive and drug-resistant strains (such as Cell-freeAMP, HydrAMP, and FoundationAMP), providing a more comprehensive perspective for studying the effects of antimicrobial peptides on different strains.

This rich and diverse dataset not only helps to reduce bias in model evaluation but also enhances the understanding of antimicrobial heterogeneity. By conducting extensive tests across multiple species and strains, our method demonstrates stronger adaptability and robustness, thereby ensuring greater reliability and practicality in real-world applications.

## 3.2 Ground Truth Assembly for the Training Set

The Ground Truth (GT) for our training data, namely the MIC values for the six classes of bacteria, was sourced from the DBAASP and PDB databases. Given that longer sequences require biologically more expensive synthesis methods compared to cheaper chemical synthesis, only peptides ranging from 6 to 30 amino acids in length were included. For the DBAASP database, a series of data cleansing algorithms were employed to obtain average MIC values for each class of DR bacteria, while peptides from the PDB database were designated as negative samples (Wang et al., 2024). The peptides consist exclusively of the 20 standard amino acids. Considering the challenge of obtaining a comprehensive dataset from existing databases, we employed the Multiple Imputation by Chained Equations (MICE) (van Buuren & Groothuis-Oudshoorn, 2011) to impute missing MIC values. This imputation method demonstrated superior results on the wet-lab dataset compared to the direct imputation of ineffective MIC values.

Table 2 presents a comparison of various datasets used for antimicrobial peptide (AMP) identification, emphasizing their contributions in terms of sequence count, species diversity, structural data availability, and data cleanliness. Notably, our proposed dataset, **DRAMPAtlas 1.0**, introduced in this work, comprises 23,673 sequences and includes structures for all entries, totaling 47,346 structures due to multiple conformations per peptide. It also covers six microbial species and has been thoroughly cleaned to ensure high data quality. Compared to other datasets like LAMPv2 (Ye et al., 2020), dbAMP 2.0 (Jhong et al., 2022), and DRAMPv3 (Shi et al., 2022), which either lack structural information or data cleaning, DRAMPAtlas 1.0 provides a more comprehensive and reliable resource for AMP research.

Furthermore, while QLAPD (Wang et al., 2024) also offers cleaned data with structures for 12,914 sequences across six species, DRAMPAtlas 1.0 significantly expands on the structural annotations and sequence count. The availability of extensive structural data in our dataset facilitates advanced modeling approaches that leverage three-dimensional information, potentially leading to more accurate AMP identification. This comprehensive dataset addresses the limitations of previous resources by providing high-quality, well-annotated data, thus serving as a valuable asset for the research community in developing and benchmarking AMP prediction models.

## 3.3 Representation and Learning of Antimicrobial Peptides

In the domain of antimicrobial peptide (AMP) identification, the representation of peptides is a pivotal factor influencing the performance of machine learning models. Different representations encapsulate varying levels of biological and chemical information, affecting the models' ability to learn discriminative features. In this study, we considered three primary types of representations for AMPs: sequence representations, structural representations, and global descriptor representations. Each representation offers unique insights into the peptides' properties, and understanding their impact on model learning is essential for advancing AMP identification.

### 3.3.1 SEQUENCE REPRESENTATION

Amino acids in the peptides were encoded using integers ranging from 1 to 20, corresponding to the 20 standard amino acids. Sequences were padded with 0 to handle varying lengths. This integer encoding was often transformed into a one-hot encoding, resulting in a binary vector representation for each amino acid residue. One-hot encoding preserves the categorical nature of amino acids and allows models to learn from the positional information within sequences.

Sequence representations focus on the primary structure of peptides, providing foundational information for models to identify motifs and patterns associated with AMPs.

### 3.3.2 STRUCTURAL REPRESENTATION

To incorporate three-dimensional structural information, we employed structural representations involving two key steps: structural prediction and structural encoding.

**Structural Prediction**    We predicted the tertiary structures of peptides using advanced algorithms:

- **AlphaFold** (Jumper et al., 2021): A state-of-the-art protein structure prediction tool utilizing multiple sequence alignments and transformer architectures to predict protein folding.
- **HelixFold** (Fang et al., 2023): A BERT-like encoder-decoder architecture that predicts structures directly from sequences without requiring homologous sequences.

Since many peptides lack sufficient homologous sequences, especially short AMPs, we adapted the input sequences for AlphaFold by repeating them to meet the algorithm's requirements. Detailed hyperparameters for structure prediction are provided in the Appendix.

**Structural Encoding**    Structural representations aim to provide models with comprehensive spatial and physicochemical information, potentially enhancing the identification of structural motifs relevant to antimicrobial activity. We encoded them using:

1. **Graph Neural Networks (GNNs)**: Each amino acid residue was represented as a node in a graph, with edges representing spatial proximity and sequential relationships. In our graph representation (Gong et al., 2023), each node features a 43-dimensional vector comprising: *1. Amino Acid Embedding* (20 dimensions): A one-hot encoding of the amino acid type. *2. Energy Information* (20 dimensions): Energy-related features pertinent to the amino acid. *3. 3D Position* (3 dimensions): Cartesian coordinates representing the spatial location of the amino acid residue. This representation captures both the structural and physicochemical properties of the peptides within the graph framework.

2. **Voxel Representation**: We encoded molecular characteristics into a three-dimensional voxel grid (Wang et al., 2024). The voxel representation had dimensions of $64 \times 64 \times 64$ with four channels: *1. Atom Weight*: Encoding the atomic mass of the amino acids. *2. Amino Acid Charge*: Representing the electric charge properties. *3. Amino Acid Category*: Categorical classification of amino acids (e.g., hydrophobic, polar). *4. Hydrophilic Information*: Indicating the hydrophilicity of amino acids.

    Voxel representations allow neural networks to process 3D spatial data, capturing intricate structural details of peptides.

### 3.3.3 GLOBAL DESCRIPTOR REPRESENTATION

Global descriptors offer aggregated properties of peptides derived from sequences or structures. Each peptide was represented by a 10-dimensional vector encompassing features such as charge, and hydrophobicity. These descriptors provide holistic insights into the peptides' biochemical properties, which can be crucial for distinguishing AMPs from non-AMPs. The detailed list of global descriptors used is available in Appendix Table 2.

### 3.3.4 LEARNING REPRESENTATIONS

Understanding the learning process from these representations is vital for optimizing model performance. We experimented with a wide array of machine learning methods to evaluate their efficacy:

Table 3: **Comparison of various deep learning methods with different AMP representations.** We chose commonly used machine learning methods for sequence, voxel, and graph representation of AMPs. We conducted a fair setting for every method. Detailed settings are provided in Appendix C. Metrics below (mean±s.d.) were tested on the 151 samples DRAMPAtlas test set across 5 folds. In an overall view, methods rely on sequence representation to perform the best, followed by graph methods and then voxel methods.

| Category | Model | AP | F1 | ACC | AUC |
|---|---|---|---|---|---|
| 3.1 seq. | SVM (Cortes & Vapnik, 1995) | $55.25_{\pm 2.64}$ | $69.25_{\pm 0.00}$ | $53.64_{\pm 0.00}$ | $54.20_{\pm 5.21}$ |
| | XGBoost (Chen & Guestrin, 2016) | $61.55_{\pm 2.10}$ | $69.25_{\pm 0.00}$ | $53.64_{\pm 0.00}$ | $65.37_{\pm 3.30}$ |
| | CatBoost (Prokhorenkova et al., 2018) | $71.33_{\pm 5.05}$ | $70.86_{\pm 0.26}$ | $57.15_{\pm 0.56}$ | $73.05_{\pm 5.10}$ |
| | MLP (Rumelhart et al., 1986) | $70.95_{\pm 4.53}$ | $68.64_{\pm 0.98}$ | $55.06_{\pm 2.72}$ | $65.34_{\pm 7.34}$ |
| | GRU (Chung et al., 2014) | $84.93_{\pm 0.30}$ | $72.98_{\pm 1.27}$ | $61.50_{\pm 2.54}$ | $87.41_{\pm 0.67}$ |
| | LSTM (Hochreiter & Schmidhuber, 1997) | $85.55_{\pm 0.25}$ | $72.73_{\pm 1.39}$ | $60.99_{\pm 2.79}$ | $88.50_{\pm 0.43}$ |
| | RNN (Elman, 1990) | $78.80_{\pm 4.44}$ | $71.54_{\pm 0.60}$ | $58.59_{\pm 1.24}$ | $82.59_{\pm 2.57}$ |
| | Transformer (Vaswani et al., 2017) | $81.81_{\pm 1.78}$ | $74.64_{\pm 2.21}$ | $69.54_{\pm 6.64}$ | $84.01_{\pm 2.51}$ |
| | Mamba (Dao & Gu, 2024) | $78.81_{\pm 1.56}$ | $75.99_{\pm 2.64}$ | $71.17_{\pm 2.57}$ | $79.34_{\pm 2.29}$ |
| 3.2.1 structure voxel | ResNet (He et al., 2016) | $73.54_{\pm 2.12}$ | $68.00_{\pm 2.01}$ | $61.26_{\pm 5.24}$ | $73.05_{\pm 1.90}$ |
| | DenseNet (Huang et al., 2017) | $76.49_{\pm 1.10}$ | $70.61_{\pm 4.38}$ | $64.42_{\pm 1.62}$ | $74.60_{\pm 1.38}$ |
| | ConvNeXt (Liu et al., 2022) | $61.22_{\pm 4.63}$ | $69.73_{\pm 0.98}$ | $55.06_{\pm 2.83}$ | $56.42_{\pm 5.24}$ |
| | ViT (Dosovitskiy et al., 2021) | $70.31_{\pm 1.31}$ | $69.49_{\pm 0.49}$ | $55.65_{\pm 4.02}$ | $70.93_{\pm 1.92}$ |
| | SwinTransformer (Liu et al., 2021) | $56.75_{\pm 0.94}$ | $69.25_{\pm 0.00}$ | $53.64_{\pm 0.00}$ | $51.83_{\pm 1.02}$ |
| 3.2.2 structure graph | GCN (Kipf & Welling, 2017) | $80.39_{\pm 2.33}$ | $73.37_{\pm 0.42}$ | $62.32_{\pm 0.85}$ | $82.20_{\pm 2.13}$ |
| | GraphSAGE (Hamilton et al., 2017) | $77.37_{\pm 0.97}$ | $73.08_{\pm 1.14}$ | $61.79_{\pm 2.37}$ | $79.63_{\pm 0.31}$ |
| | GAT (Veličković et al., 2018) | $80.33_{\pm 1.93}$ | $76.17_{\pm 0.91}$ | $67.90_{\pm 1.91}$ | $82.87_{\pm 1.33}$ |
| | GIN (Xu et al., 2019) | $79.21_{\pm 1.60}$ | $76.20_{\pm 1.35}$ | $69.71_{\pm 2.75}$ | $80.32_{\pm 1.99}$ |
| | GATv2 (Brody et al., 2022) | $78.69_{\pm 1.57}$ | $73.62_{\pm 1.29}$ | $62.85_{\pm 2.48}$ | $81.51_{\pm 2.20}$ |

**1. Conventional Machine Learning Methods**: Models such as Support Vector Machines (SVM), XGBoost, and CatBoost were utilized, leveraging their robustness in handling tabular and sequential data. **2. Deep Learning Architectures**: We explored modern architectures including Multi-layer Perceptrons (MLP), Recurrent Neural Networks (RNN), Long Short-Term Memory networks (LSTM), Graph Neural Networks (GNN), and convolutional networks suitable for voxel data.

We conducted in-depth analyses of the predicted structures from different algorithms to understand their impact on model learning. Additionally, recognizing the significant issue of data imbalance in AMP identification (stemming from the relatively small number of known AMPs compared to non-AMPs), we performed imbalance-aware loss functions to mitigate the effects of skewed class distributions. Detailed analyses and discussions of these representations and learning methods are presented in Section 4. Our investigations underscore the importance of selecting appropriate representations and learning algorithms to enhance the predictive performance of AMP identification models. By benchmarking these approaches, we aim to provide insights that facilitate the development of more accurate and generalizable models in the field of antimicrobial peptide research.

# 4 METHOD AND EXPERIMENT

## 4.1 EVALUATION METRICS AND IMPLEMENTATION

To evaluate our method, we employ four key metrics: Precision, F1-score, Accuracy, and the Area Under the Receiver Operating Characteristic Curve (AUC). These metrics offer a comprehensive assessment of model performance across various dimensions, with Precision measuring the accuracy of positive predictions, F1-score providing a balance between Precision and Recall, Accuracy reflecting overall correctness, and AUC evaluating the model's ability to distinguish between classes under varying threshold settings.

For data pre-processing and partition, peptides with MIC values less than 128 were considered active, and those with values greater or equal to 128 were deemed inactive, forming the labels for our multi-label classification task. To prevent data leakage, we compared the similarity of each peptide in the test set with those in the training set, excluding any training peptides with more than 30% similarity to those in the test set, remaining 13k samples for further process. The remaining training

Table 4: **Comparison on different structure source and model scale.** We utilize structural predictions from AlphaFold and HelixFold to construct voxel, graph, and global descriptor inputs for ML models. AlphaFold, being the more accurate of the two prediction methods, demonstrates the significance of structural accuracy on performance. For model scaling, ResNet is selected for its extensibility, which illustrates the influence of a model's width (@ indicates the channel number) and depth (number of layers) on performance. Detailed settings are shown in Appendix D.3.

| | Structure voxel | Structure graph | AP | F1 | ACC | AUC |
|---|---|---|---|---|---|---|
| 4.1 Structure Prediction | Alphafold | - | $73.54_{\pm2.12}$ | $68.00_{\pm2.01}$ | $61.26_{\pm5.24}$ | $73.05_{\pm1.90}$ |
| | Helixfold | - | $73.38_{\pm0.89}$ | $69.22_{\pm1.86}$ | $59.23_{\pm3.03}$ | $70.02_{\pm1.39}$ |
| | - | Alphafold2 | $80.33_{\pm1.93}$ | $76.17_{\pm0.91}$ | $67.90_{\pm1.91}$ | $82.87_{\pm1.33}$ |
| | - | Helixfold | $81.06_{\pm1.90}$ | $73.61_{\pm0.77}$ | $63.47_{\pm1.26}$ | $78.95_{\pm2.86}$ |
| 4.2 Global Descriptor | Sequence (10d) | | $82.81_{\pm1.52}$ | $75.03_{\pm0.80}$ | $65.52_{\pm1.51}$ | $85.72_{\pm1.46}$ |
| | Alphafold (3d+7d from Seq.) | | $80.71_{\pm1.00}$ | $75.29_{\pm0.74}$ | $66.05_{\pm1.42}$ | $83.10_{\pm1.37}$ |
| | Helixfold (3d+7d from Seq.) | | $81.13_{\pm1.29}$ | $75.03_{\pm0.61}$ | $65.52_{\pm1.17}$ | $83.52_{\pm0.97}$ |
| 4.3 Scale up | ResNet34@16 | | $73.54_{\pm2.12}$ | $68.00_{\pm2.01}$ | $61.26_{\pm5.24}$ | $73.05_{\pm1.90}$ |
| | ResNet34@32 | | $76.71_{\pm1.76}$ | $72.04_{\pm0.53}$ | $63.07_{\pm2.03}$ | $75.05_{\pm2.12}$ |
| | ResNet50@16 | | $74.15_{\pm4.66}$ | $70.91_{\pm1.41}$ | $59.65_{\pm4.09}$ | $72.99_{\pm5.14}$ |
| | ResNet50@32 | | $73.88_{\pm4.23}$ | $71.38_{\pm1.97}$ | $62.38_{\pm2.30}$ | $72.66_{\pm4.76}$ |

set peptides were divided into five folds for cross-validation, with the best-performing samples from each fold evaluated on the test set.

**Computational Infrastructure and Software Setup**: Our computational setup includes an Intel Core i9-14900K CPU and two NVIDIA 4090D GPUs, each with 24GB of VRAM, complemented by 256GB of RAM. System storage is managed by a 4TB Samsung 990 PRO SSD. The platform runs on Ubuntu 22.04, with NVIDIA Driver Version 550.107.02 and CUDA Version 12.4. Software configurations are as follows: PyTorch version 2.2.0 (Paszke et al., 2019), torch-geometric version 2.5.3 (Fey & Lenssen, 2019), and Python version 3.10.0. Peptide sequence global attributes were processed using BioPython version 1.78 (Cock et al., 2009), and evaluation metrics were computed using torchmetrics version 1.4.0 (Detlefsen et al., 2022). For detailed hyperparameter settings, please refer to Supplementary Material Table 2.

## 4.2 ANALYSIS ON THE REPRESENTATION OF AMPS

Table 3 presents a comprehensive comparative analysis of various feature representation methods for antimicrobial peptides (AMPs), highlighting the performance metrics across different deep-learning models. The table underscores the significant impact that feature representation has on the predictive capabilities of machine learning models in the context of AMP classification. Notably, it is observed that simple sequence-based representation methods outperform more complex representations that incorporate spatial, energy, and other structural information. This is particularly surprising, as one might intuitively expect richer representations to yield better performance due to the additional information they encapsulate.

Sequence-based models such as LSTM and Transformer demonstrate superior performance, achieving average precision (AP) scores of 85.55% and 81.81% respectively. These models capitalize on the sequential nature of amino acid sequences, effectively capturing the essential patterns and dependencies necessary for accurate classification. In contrast, voxel-based methods, despite their computational intensity and the inclusion of three-dimensional structural data, lag behind in performance. For instance, the ResNet model with voxel representation attains an AP of only 73.54%, significantly lower than its sequence-based counterparts. This underperformance may be attributed to issues in the construction of voxel-based inputs or potential challenges in the imputation of labels during training.

Moreover, within each representation category, most models exhibit comparable performance levels, with few outliers significantly underperforming. Graph-based methods, which encode structural information in the form of graphs, also show promising results. The GAT model, for example, achieves an AP of 80.33% and a relatively high F1 score of 76.17%, suggesting that attention mechanisms in

Table 5: **Analysis on the fusion of different representations.** The networks used to extract the feature are shown in the left part. We adopt the resnet34 as the voxel feature extracting backbone, while GAT is used to extract the feature from graph representations.

| Sequence | Structure | Descriptor | AP | F1 | ACC | AUC |
|---|---|---|---|---|---|---|
| LSTM | - | - | $85.55_{\pm 0.25}$ | $72.73_{\pm 1.39}$ | $60.99_{\pm 2.79}$ | $88.50_{\pm 0.43}$ |
| - | - | MLP | $82.81_{\pm 1.52}$ | $75.03_{\pm 0.80}$ | $65.52_{\pm 1.51}$ | $85.72_{\pm 1.46}$ |
| LSTM | - | MLP | $82.15_{\pm 1.82}$ | $76.40_{\pm 2.03}$ | $67.99_{\pm 3.82}$ | $83.89_{\pm 2.31}$ |
| LSTM | ResNet | - | $77.88_{\pm 2.67}$ | $73.55_{\pm 1.95}$ | $64.53_{\pm 3.19}$ | $76.68_{\pm 2.63}$ |
| LSTM | GAT | - | $76.85_{\pm 3.81}$ | $73.64_{\pm 0.46}$ | $62.85_{\pm 0.91}$ | $80.05_{\pm 2.57}$ |
| - | ResNet | MLP | $78.70_{\pm 1.15}$ | $69.50_{\pm 2.41}$ | $67.15_{\pm 1.75}$ | $78.04_{\pm 1.62}$ |
| - | GAT | MLP | $79.23_{\pm 1.17}$ | $73.74_{\pm 1.97}$ | $62.94_{\pm 3.78}$ | $81.17_{\pm 1.95}$ |
| LSTM | ResNet | MLP | $82.02_{\pm 1.20}$ | $74.66_{\pm 2.00}$ | $66.98_{\pm 5.21}$ | $81.05_{\pm 1.52}$ |
| LSTM | GAT | MLP | $78.51_{\pm 1.98}$ | $72.77_{\pm 0.83}$ | $61.10_{\pm 1.68}$ | $80.57_{\pm 2.71}$ |

graph networks effectively capture important structural relationships. These findings indicate that while complex representations do not necessarily guarantee better performance, certain architectures like LSTM, ResNet, and GAT models interact more effectively with specific feature encoding schemes. Further investigation into these top-performing models may provide deeper insights into optimizing representation methods for AMP classification. Besides, the sequenced-based methods generally outperform the two other structure representation methods. ***Finding 1: Sequence information is important due to its accuracy.***

Table 4 presents a comprehensive comparison of different structure sources and model scales in the context of antimicrobial peptide (AMP) identification. In Table 4.1, we evaluate the impact of using structural predictions from AlphaFold and HelixFold to construct voxel and graph inputs for machine learning models. The results indicate that graph-based representations consistently outperform voxel-based ones. Specifically, when utilizing graph representations derived from AlphaFold predictions, the model achieves an Average Precision (AP) of 80.33% and an AUC of 82.87%. This is a significant improvement over voxel representations, which attain an AP of 73.54% and an AUC of 73.05% using AlphaFold. The superior performance of graph representations suggests that graph-based inputs may better capture the structural nuances crucial for AMP activity prediction. Additionally, while both AlphaFold and HelixFold are effective for graph construction, AlphaFold slightly outperforms HelixFold in terms of AP and AUC, highlighting ***Finding 2: the accurate structural predictions will lead to better result***, as the AlphaFold provides more accurate structure prediction result than HelixFold.

Table 4.2 shows the effect of different types of global descriptors and their impact on the identification performance. The results show that the global information like alpha-sheet fraction from sequence is slightly better than the information from structure, which might be due to the inaccurate predicted AMP's structure. ***Finding 3: global information from sequence is more beneficial for AMP identification than from structure.***

In Table 4.3, we examine how scaling the model's width and depth influences performance by experimenting with different configurations of ResNet models (He et al., 2016). Increasing the network width from 16 to 32 channels yields noticeable improvements; for instance, ResNet34@32 achieves an AP of 76.71% compared to 73.54% with ResNet34@16. This enhancement suggests that wider networks can capture more discriminative features essential for accurate AMP identification. Conversely, scaling the model depth from ResNet34 to ResNet50 does not consistently lead to better performance. In some cases, deeper models like ResNet50@32 show marginal decreases in AP and AUC compared to their shallower counterparts. This observation may indicate that beyond a certain depth, the benefits of additional layers diminish, possibly due to overfitting or vanishing gradient issues, as discussed in He et al. (2016). Overall, the results underscore that model width plays a more critical role than depth in this specific application, and that carefully selecting the structural representation and model architecture is crucial for optimizing AMP identification models. ***Finding 4: properly scale up the channel number benefits the performance.***

Table 6: **Comparison on different rebalancing method.** We choose the LSTM+MLP ablation in Table 5 as the baseline. DistributedBalancedLoss has 4 reweight functions implemented: Inverse (Inv), Squareroot inverse (Sqrt inv), Rebalance, and Class-balance (CB).

| Methods | | AP | F1 | ACC | AUC |
|---|---|---|---|---|---|
| No Rebalancing | | $82.15_{\pm1.82}$ | $76.40_{\pm2.03}$ | $67.99_{\pm3.82}$ | $83.89_{\pm2.31}$ |
| DistributionBalancedLoss (Wu et al., 2020) | Inv | $82.37_{\pm2.85}$ | $76.62_{\pm1.32}$ | $68.92_{\pm3.00}$ | $84.14_{\pm3.78}$ |
| | Sqrt inv | $80.30_{\pm1.89}$ | $77.08_{\pm1.06}$ | $70.24_{\pm2.61}$ | $81.93_{\pm1.56}$ |
| | Rebalance | $80.13_{\pm1.65}$ | $77.19_{\pm0.80}$ | $70.18_{\pm1.83}$ | $82.05_{\pm1.86}$ |
| | CB | $38.08_{\pm0.00}$ | $0.00_{\pm0.00}$ | $46.36_{\pm0.00}$ | $18.71_{\pm0.00}$ |
| ZLPR (Su et al., 2022) | | $82.06_{\pm0.83}$ | $75.85_{\pm1.72}$ | $67.04_{\pm3.25}$ | $84.37_{\pm1.21}$ |
| WeightedBCE (Rezaei-Dastjerdehei et al., 2020) | | $82.25_{\pm1.67}$ | $78.28_{\pm1.64}$ | $71.83_{\pm3.25}$ | $84.52_{\pm1.98}$ |

### 4.3 ANALYSIS ON THE LEARNING PROCESS OF AMPs

Table 5 presents an analysis on the fusion of different representations for antimicrobial peptide (AMP) identification during the training process. Models combining various feature extraction networks for sequence (**LSTM**), structure (**ResNet** or **GAT**), and descriptors (**MLP**) are evaluated using metrics such as Average Precision (**AP**), F1 score (**F1**), Accuracy (**ACC**), and Area Under the ROC Curve (**AUC**). The results show that using only sequence information with an LSTM network achieves the highest AP and AUC scores of 85.55% and 88.50%, respectively, indicating strong predictive power when modeling sequential data alone.

Furthermore, combining sequence features with descriptors generally leads to competitive performance. For example, the combination of LSTM and MLP yields an F1 score of 76.40% and an ACC of 67.99%, showing improvement over using descriptors alone. However, incorporating structural information through ResNet or GAT does not consistently enhance performance, which may be due to the complexity of structural data or potential overfitting. Overall, the results suggest that sequence-based models, possibly augmented with descriptor information, are effective for AMP prediction, while the benefits of including structural representations require further investigation. The potential reason is that the predicted structure of the AMPs is not that accurate. ***Finding 5: fusion of different types of representation can be useful when we select proper structure, but the inaccuracy of structure information still hurts.***

Table 6 compares different rebalancing methods applied to the LSTM+MLP baseline model from Table 5. The methods evaluated include various weighting schemes from DistributionBalancedLoss (Wu et al., 2020), namely Inverse (Inv), Squareroot inverse (Sqrt inv), Rebalance, and Class-Balance (CB), as well as ZLPR (Su et al., 2022) and Weighted Cross-Entropy (WeightedCE) (Rezaei-Dastjerdehei et al., 2020). The results indicate that rebalancing methods generally improve performance metrics such as Average Precision (**AP**), F1 score, Accuracy (**ACC**), and Area Under the ROC Curve (**AUC**) compared to no rebalancing. Specifically, the WeightedCE method achieves the highest F1 score (78.28%) and ACC (71.83%), suggesting that weighted loss functions effectively address the class imbalance in the dataset. From the above analysis, we can draw the conclusion that addressing the data imbalance is important in boosting the identification performance. ***Finding 6: re-balancing is useful in AMPs identification.***

## 5 CONCLUSION AND DISCUSSION

Our contributions are twofold and aim to mitigate the limitations currently faced by the AI-driven discovery of AMPs. Firstly, the creation of **DRAMPAtlas 1.0** represents a pivotal advancement in providing a standardized dataset that includes both public data and new wet-lab experimental results. This dataset not only enriches the available data with comprehensive activity and toxicity information across six types of DR bacteria but also integrates these with 3D structural data of each peptide. Secondly, our extensive experiments utilizing both voxel and graph representations of AMPs, in conjunction with sequence data, have provided valuable insights. We believe These interesting findings will provide new insights into the research community.

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

CONTENTS

## A  APPENDIX OF WET LAB EXPERIMENTS

All bacterial strains were initially streaked onto Nutrient Broth (NB) agar medium and incubated overnight at 37°C to ensure proper growth. For antimicrobial activity assessments, Minimum Inhibitory Concentration (MIC) assays of antimicrobial peptides were conducted using Mueller Hinton broth in accordance with Clinical and Laboratory Standards Institute (CLSI) guidelines. The bacterial colonies were first suspended in a saline solution, and the turbidity was adjusted to a McFarland standard of 0.5 to achieve a bacterial concentration of $10^8$ CFU/mL, followed by a 100-fold dilution for the inhibition test. Subsequently, 50μL of the bacterial suspension at $1 \times 10^6$ CFU/mL were incubated with an equal volume of different concentrations of peptide solutions (serial 2-fold dilutions in Mueller-Hinton Broth from Hopebio). After incubation for 18–20 hours at 37°C, the MIC was recorded as the lowest concentration at which no obvious bacterial growth was observed. Fifty microliters (50μL) of the mixture without observed bacteria were added to Mueller-Hinton (MH) agar plates. After further incubation for 18 hours at 37°C, the Minimum Bactericidal Concentration (MBC) was recorded as the lowest concentration with no bacterial growth. All experiments were performed in biological triplicates to ensure reproducibility and reliability of the results. The detailed bacterial species used in our studies are shown in Tab. 1.

Table 1: Drug-Resistant bacterial strains used in the study. The number without prefix indicates the DR bacterial from clinical environment.

| A. baumannii | P. aeruginosa | E. coli | K. pneumoniae | S. enterica | S. aureus |
|---|---|---|---|---|---|
| 102 | 112 | 207 | 201 | ATCC14028 | 101 |
| 104 | 116 | 208 | 209 | CMCC50071 | 103 |
| 106 | 1162 | 210 | 212 | | 1032 |
| | | | | | 129 |

## B   APPENDIX OF STRUCTURE PREDICTION DETAILS

In this study, we employed advanced computational frameworks to predict the three-dimensional structures of antimicrobial peptides (AMPs). Specifically, we utilized ColabFold (Mirdita et al., 2022), which integrates AlphaFold2 (Jumper et al., 2021), and HelixFold (Fang et al., 2023) for structure prediction. Additionally, we refined the predicted structures using Rosetta (Leaver-Fay et al., 2011) to enhance structural accuracy.

### B.1   STRUCTURE PREDICTION USING COLABFOLD AND ALPHAFOLD2

We adopted the ColabFold framework, which leverages the AlphaFold2 algorithm for protein structure prediction. ColabFold accelerates the prediction process through optimized implementations and allows for easy execution on cloud-based platforms. The detailed command used for structure prediction is as follows:

```
colabfold_batch ./a3ms/ ./results/ --amber --use-gpu-relax
                --num-relax 1 --num-models 3 --model-order 3,4,5
```

In this command:

- `./a3ms/` specifies the directory containing input multiple sequence alignments (MSAs) in A3M format.
- `./results/` designates the directory where predicted structures and related outputs are saved.
- `--amber` enables relaxation using the AMBER force field for improved structural refinement.
- `--use-gpu-relax` allows the relaxation to be performed using GPU acceleration, enhancing computational efficiency.
- `--num-relax 1` sets the number to 1 for how many of the top ranked structures to relax.
- `--num-models 3` specifies that three models will be predicted for each input sequence.
- `--model-order 3,4,5` indicates that AlphaFold2 models 3, 4, and 5 will be used in the prediction process.

The use of the AMBER force field and multiple models aims to improve the accuracy of the predicted structures by providing optimized energy states and accounting for model variability (Jumper et al., 2021).

### B.2   STRUCTURE PREDICTION USING HELIXFOLD AND REFINEMENT WITH ROSETTA

For additional validation, we employed HelixFold (Fang et al., 2023), a protein structure prediction framework designed for high-throughput and high-accuracy modeling. We used HelixFold to predict structures for the AMPs, focusing on capturing intricate structural features that might not be fully resolved by other methods.

Post-prediction, we refined the structures using the Rosetta software suite (Leaver-Fay et al., 2011), a widely used tool for protein structure prediction and refinement. Rosetta facilitates energy minimization and conformational sampling, improving stereochemical properties and reducing potential errors in side-chain orientations and backbone conformations. The refinement process involves:

1. Relaxing the protein structure to find the lowest energy conformation.
2. Fine-tuning side-chain packing to enhance hydrophobic interactions and eliminate steric clashes.
3. Optimizing hydrogen-bond networks to stabilize secondary and tertiary structures.

The integration of HelixFold and Rosetta refinement ensures that the predicted structures are as accurate and reliable as possible for downstream analyses.

## C   APPENDIX OF DESCRIPTOR DEFINITION

The specific definition of each digit in the descriptor vector is shown in the Tab.2. Among them, Alpha helix (Index 4), Beta sheet (Index 5), and Turn helix (Index 6) can be extracted either from sequence with Biopython or from structure predicted by AlphaFold or HelixFold with MDTraj. All other properties are extracted from the sequence with Biopython.

Table 2: Definition of descriptor vector

| Dim Index | Property | Dim Index | Property | Dim Index | Property |
|---|---|---|---|---|---|
| 0 | Gravy | 4 | Alpha helix | 8 | Isoelectric point |
| 1 | Aliphatic index | 5 | Beta sheet | 9 | Charge density |
| 2 | Aromaticity | 6 | Turn helix | | |
| 3 | Instability index | 7 | Charge at pH 7 | | |

## D   APPENDIX OF MACHINE LEARNING EXPERIMENTS

### D.1   COMMON DEFAULT SETTINGS

If not specified, all the setting in Tab.3 is used.

Table 3: Common experiment settings

| | | | |
|---|---|---|---|
| Optimizer | AdamW | Batch size | 256 |
| Weight decay | 0.01 | Base hidden dims (Seq.) | 128 |
| Learning rate | 0.001 | Base hidden dims (Voxel) | 16 |
| Loss function | BCEWithLogitLoss | Base hidden dims (Graph) | 128 |
| Epochs each fold | 50 | Peptide structure source | AlphaFold |

### D.2   SETTINGS OF TABLE 3

For sequence models, bidirectional forward is enabled if possible, layer number is set to 2. Additionally for MLP, 3 layers and a dropout rate of 0.3 are specified; for the transformer, it has 4 attention heads.

For voxel models, the DenseNet growth rate is set to 16, ViT patch size is (8, 8, 8), SwinTransformer patch size is (4, 4, 4), and window size (2, 2, 2)

For graph models, 2 layers and a dropout rate of 0.3 is set.

### D.3   SETTINGS OF TABLE 4

In Table 4.1, the voxel method uses ResNet34@16, graph method uses GAT@128. In Table 4.2, a 3-layer MLP@128 is used to encode the descriptor.

# E APPENDIX OF VOXEL AND GRAPH FEATURES ABLATION

We conducted ablation experiments on features encoded to the peptide representations. Most of the experiment settings are shared with Appendix C, voxel-based experiments use model ResNet34@16, and graph-based experiments use model GAT@128. Detailed results are shown in Tab.4

The ablation study reveals several key findings regarding the impact of feature removal on performance metrics. For voxel features, the exclusion of charge yields the highest AP at 76.36 and AUC at 74.26. Conversely, removing the hydrophilic type results in the lowest performance across metrics, underscoring its critical role in maintaining accuracy and AUC. In the context of graph features, the absence of energy leads to the highest AP at 84.74 and AUC at 88.23. Meanwhile, removing amino acid information significantly diminishes performance, highlighting its importance for sustaining metric scores.

Table 4: Impact of feature removal on voxel and graph Representations

| Ablation type | Info removed | AP | F1 | ACC | AUC |
|---|---|---|---|---|---|
| Voxel | No remove | $73.54_{\pm 2.12}$ | $68.00_{\pm 2.01}$ | $61.26_{\pm 5.24}$ | $73.05_{\pm 1.90}$ |
| | Atom weight | $74.31_{\pm 3.16}$ | $71.06_{\pm 1.13}$ | $62.38_{\pm 2.03}$ | $72.67_{\pm 3.30}$ |
| | AA Hydrophilic type | $69.47_{\pm 4.04}$ | $66.89_{\pm 2.21}$ | $57.92_{\pm 2.57}$ | $66.09_{\pm 4.74}$ |
| | AA Charge | $76.36_{\pm 2.46}$ | $70.07_{\pm 0.87}$ | $61.08_{\pm 1.89}$ | $74.26_{\pm 2.79}$ |
| | AA Category | $73.30_{\pm 0.82}$ | $70.17_{\pm 1.45}$ | $61.41_{\pm 2.39}$ | $71.50_{\pm 1.17}$ |
| Graph | No remove | $80.33_{\pm 1.93}$ | $76.17_{\pm 0.91}$ | $67.90_{\pm 1.91}$ | $82.87_{\pm 1.33}$ |
| | AA Embedding | $64.03_{\pm 1.25}$ | $70.83_{\pm 0.31}$ | $57.09_{\pm 0.65}$ | $65.35_{\pm 0.88}$ |
| | Energy | $84.74_{\pm 0.61}$ | $75.69_{\pm 0.51}$ | $66.78_{\pm 0.94}$ | $88.23_{\pm 0.55}$ |
| | Position | $80.84_{\pm 1.45}$ | $75.81_{\pm 3.27}$ | $66.91_{\pm 6.14}$ | $83.29_{\pm 1.92}$ |

