# OpenReview forum: "Benchmarking Antimicrobial Peptide Identification with Sequence and Structure Representation"
_ICLR.cc/2025/Conference — ICLR 2025 Conference Withdrawn Submission_

### Official Review · Reviewer_FQuT · 2024-10-21

**Soundness:** 2
**Presentation:** 3
**Contribution:** 2
**Rating:** 3
**Confidence:** 4

**Summary:**

​The paper constructs the DRAMPAtlas 1.0, which contains the training set collected from the public, and the testing set derived from wet lab experiments. It includes extensive experiments for AMP identification by modeling the 3D structure as voxels or graphs, along with its sequence information or independently using either structure or sequence. ​Several intriguing findings are presented in the paper.

**Strengths:**

​1. The significance of this study in the field of antimicrobial peptide identification and development is substantial.​ By creating the DRAMPAtlas 1.0 dataset, the authors have addressed the issues related to data quality and diversity that are prevalent in existing  AMP databases.

2. Additionally, the paper presents several intriguing findings that warrant further attention.

**Weaknesses:**

​1. The technical novelty appears to be limited.​ While the authors have introduced a new dataset and method, the discussion surrounding the innovation in the paper and the comparison with existing studies is relatively weak.

2. The baseline method seems outdated. A comparison of the proposed research method against other existing antimicrobial peptide identification methods in the literature would be beneficial. A more thorough quantitative and qualitative comparative analysis could clarify the innovation and advancement of this work.

3. Regarding the DRAMPAtlas 1.0 dataset, the independent validation experiments for the new dataset presented in the paper are insufficient. It would be advantageous for the authors to perform more rigorous cross-validation to confirm the stability and generalization capability of the model.

4. Providing specific case studies or examples would not only enrich the findings but also guide future improvements in the model.

5. It is advisable to visualize the experimental data alongside the predicted antimicrobial peptide molecules and their properties. This approach will facilitate a more convenient comparison of the differences between the various datasets.
6. Reviews of computer science papers often recommend the provision of anonymous code, accompanied by straightforward and easy-to-test data. Additionally, it is suggested to include a Jupyter Notebook, which enables readers to quickly and intuitively grasp the workings of the method presented.

**Questions:**

How do the authors determine the importance of features in the model? Can feature importance analysis be included to provide some insights into which data features have the greatest impact on performance?

---

### Official Review · Reviewer_xhDV · 2024-10-30

**Soundness:** 3
**Presentation:** 1
**Contribution:** 2
**Rating:** 3
**Confidence:** 4

**Summary:**

This paper presents a new dataset and an accompanying empirical investigation of methods for predicting antimicrobial peptides (AMPs) from sequence or structure.  The authors produced a new dataset, which they use to systematically investigate a series of questions related to input representations, model architectures, etc.  Some of the conclusion are (1) sequence-based representations work better than sequence+structure representations (line 416), (2) choice of model type does not seem to have a particularly large impact (line 429), (3) graph-based representations outperform voxel-based representations (line 461), in a ResNet architecture, adding more channels is beneficial (line 485), and (4) weighted loss functions are beneficial in this setting (line 522).  Overall, I think the paper needs to be edited for clarity, particular in the introductory sections.

The overall contributions are two-fold.  First, the authors have conducted wet lab experiments to measure the antimicrobial properties of a family containing 150 AMPs relative to six different types of bacteria.  Second, they conducted empirical investigation to test the hypothesis that incorporating alphafold predicted structures can improve the ability to detect AMPs.  They also investigate various learning strategies, including different neural net architectures, feature fusion methods, and re-balancing strategies.

As far as I can tell, the experiments here were conducted rigorously, but as evidenced by my questions below, I had trouble understanding some of what was done, so I can't be certain about this.

**Strengths:**

The empirical investigations are fairly ambitious and are reasonably well described.

**Weaknesses:**

One of the surprising results is that sequence-based methods outperform structure-based methods (line 417).  I agree with the subsequent sentence that says "This underperformance may be attributed to issues in the construction of voxel-based inputs or potential challenges in the imputation of labels during training." Overall, the ambitious study design makes it difficult to follow up on these kinds of ambiguities in detail.

Regarding presentation, I was immediately struck by the fact that the first sentence uses an adjective ("microbial") as a noun ("microbe").  This usage continues thereafter.  Given how critical microbes are to this paper, this seems like a really strange error to make.  Grammatical problems continue throughout, including sentences that are incorrectly constructed (e.g., the fourth sentence in the abstract).  Trying to enumerate all of the problems here is not really feasible, but suffice to say that the use of English is poor enough to frequently impede understanding of the paper.  Incidentally, I did notice that the English improved dramatically in later portions of the paper (e.g., section 4.2).

**Questions:**

I don't understand why the text talks about testing 150 AMPs (or, in one place, 151) and Table 1 says that your dataset contains 22k sequences.  Can you explain how these two numbers are related?

Where is the evidence that your imputation strategy works well?  Please provide quantitative evidence of the imputation strategy's effectiveness, such as cross-validation results or comparisons with non-imputed data. Why not just provide the actual, unimputed data instead?  Please explain your rationale for working with imputed data rather than working with incomplete data.

I can't understand how the MIC values that you measure (line 207) relate to the MIC values that you download from the database (line 236).  Can you explain this?

The text says you measured precision (line 368), but the Table 3 header says "AP."  Did you actually measure average precision?

Why is 128 the correct MIC threshold to use?

---

### Official Review · Reviewer_YArp · 2024-11-01

**Soundness:** 2
**Presentation:** 2
**Contribution:** 2
**Rating:** 3
**Confidence:** 4

**Summary:**

AMP identification is a very important problem in bioinformatics. This study collected a new dataset DRAMPAtlas 1.0 for AMPs and evaluated the feature representations for AMP prediction.

**Strengths:**

This study collected a new dataset DRAMPAtlas 1.0 for AMPs and evaluated the feature representations for AMP prediction. The authors also conducted web lab experiments to verify the findings.

**Weaknesses:**

The motivation and innovation are not clear, the biological insight is missing.

**Questions:**

1. What are the major differences between this study with another review & benchmark study (Xu et al, 2021, Brief Bioinformatics, bbab083)?
2. The authors are suggested to provide some case studies to show the findings related to some biological insights.
3. It would be better to provide some examples of how to interpret the voxel features in the biological context.
4. Did the authors obtain any conclusions that are inconsistent with the previous benchmarks (e.g. Xu et al, 2021, Brief Bioinformatics, bbab083)?
5. The format of the references should be consistent.
6. The source codes and datasets should be publicly available.

---

### Official Review · Reviewer_3Nuk · 2024-11-02

**Soundness:** 3
**Presentation:** 3
**Contribution:** 2
**Rating:** 5
**Confidence:** 3

**Summary:**

The authors present work that aims to further the development of anti-microbial peptides (AMPs) for use in fighting drug-resistant (DR) bacteria. They are motivated by the use of machine learning methods to predict whether a given AMP is an effective treatment for a given bacterium. The ground truth for such methods are experiments which measure the effectiveness of a given AMP on a given bacterial strain, measured as its minimum inhibitory concentration (MIC). Towards this end, they present DRAMPAtlas 1.0, a benchmark for such algorithms. This benchmark consists of a combination of publicly-available data along with a set of new experiments. They furthermore a number of existing machine learning methods on this benchmark, including evaluating various combinations of model and input representation.

**Strengths:**

The paper presents a new benchmark for AMP prediction. This includes a new data set, which is rare for ML papers. The setup of the evaluation seems reasonable.

**Weaknesses:**

Soundness: See question below regarding the inclusion of existing AMP prediction methods.

Contribution: The medical/biological contribution of a new AMP data set seems quite large, although I am not very familiar with this field. However, the methodological contribution is quite small. Since ICLR focuses on methods, I must give a low rating for contribution.

Presentation: The article has may grammatical and spelling errors, which makes it hard to follow in places.

**Questions:**

Why is a novel set of experiments being submitted to ICLR? As I see it, this would be of much greater interest at an AMP venue.

There are many previously-proposed methods for AMP prediction, as cited in the paper. I understand that these methods use the models tested in e.g. Table 3, but are the models tested exactly those that were previously proposed? If not, they should be included, as e.g. the choice of hyperparameters can make a big difference.

L243: "This imputation method demonstrated superior results on the wet-lab dataset compared to the direct imputation of ineffective MIC values." This seems to represent leakage between the train and test sets, does it not?

---

### Note · Authors · 2024-11-13

I have read and agree with the venue's withdrawal policy on behalf of myself and my co-authors.